# Gut Microbiota Crosstalk with Resident Macrophages and Their Role in Invasive Amebic Colitis and Giardiasis—Review

**DOI:** 10.3390/microorganisms11051203

**Published:** 2023-05-04

**Authors:** Panagiota Xaplanteri, Nikiforos Rodis, Charalampos Potsios

**Affiliations:** 1Department of Microbiology, General Hospital of Eastern Achaia, 25001 Kalavrita, Greece; 2Department of Surgery, University General Hospital of Patras, 26332 Patras, Greece; rodis.nikiforos@gmail.com; 3Department of Internal Medicine, University General Hospital of Patras, 26504 Patras, Greece; charpotsios@gmail.com

**Keywords:** gut microbiota, macrophages, intestinal protozoan parasites, inflammasome

## Abstract

The innate immune response is highly dependent on the action of macrophages. They are abundant in the intestine subepithelial lamina propria of the mucosa, where they deploy multiple tasks and play a critical role. The balance between the gut microbiota and M2 macrophages is critical for gut health and homeostasis. Gut microbiota has the power to change macrophage phenotype and replenish the resident macrophage niche during and post infection. As far as the extracellular enteric parasitic infections invasive amebic colitis and giardiasis are concerned, a change of macrophages phenotype to a pro-inflammatory state is dependent on direct contact of the protozoan parasites with host cells. Macrophages induce strong pro-inflammatory response by inflammasome activation and secretion of interleukin IL-1β. Inflammasomes play a key role in the response to cellular stress and microbe attacks. The balance between gut mucosal homeostasis and infection is dependent on the crosstalk between microbiota and resident macrophages. Parasitic infections involve NLRP1 and NLRP3 inflammasome activation. For *Entamoeba histolytica* and *Giardia duodenalis* infections, inflammasome NLRP3 activation is crucial to promote the host defenses. More studies are needed to further elucidate possible therapeutic and protective strategies against these protozoan enteric parasites’ invasive infections in humans.

## 1. Introduction

The innate immune response is highly dependent on the action of macrophages [1]. These are abundant in the intestine subepithelial lamina propria of the mucosa, where they deploy multiple tasks and play a critical role. Pro-inflammatory sentinel macrophages express the M1 phenotype. Macrophages that express the M2 phenotype promote wound healing, through induction of transforming growth factor β1 (TGF-β1). TGF-β1 recruits tissue fibroblasts to boost tissue repair [2]. Macrophages are also positioned in the muscularis externa near sympathetic nervous system neurons and they seem to influence enteric motility [3]. Intestine macrophages have the unique characteristic of a continuous steady supply from Ly6C+ blood monocytes even in the absence of inflammation [3].

These monocytes, after migration to gut mucosa, differentiate into CX3CR1hi macrophages which produce PGE2 and interleukin-10 (IL-10), maintaining anti-inflammatory properties and maintaining the integrity and homeostasis of the enteric mucosa [3].

In the absence of inflammation, the gut mucosal surface forms a barrier between the intestinal epithelium and enteric microbiota [4]. This dense mucus barrier is constructed by the mucin MUC2. MUC2 has many properties and functions; it acts as a shield and educates intestinal dendritic cells to retain their anti-inflammatory properties in the absence of inflammation [4]. Gut antigen-presenting cells and microbiota cooperate to maintain the delicate balance between infection and intestinal homeostasis [5]. Alterations in the balance of symbiosis between the gut microbiota and enteric epithelium are described to act in favor of protozoan invasive infections [6].

Macrophages induce strong pro-inflammatory responses through inflammasome activation and secretion of interleukin IL-1b (IL-1b) [1]. Inflammasomes are cytosolic structures that are part of the innate immunity [7]. Inflammasomes were first described in 2002 as cytosolic complexes composed of a sensor molecule (usually a Nod-like receptor, NLR), an adaptor protein (usually apoptosis-associated speck-like protein containing a CARD, ASC), a highly conserved NACHT domain with nucleoside-triphosphatase (NTPase) activity, and an inactive cysteine protease caspase-1 (CASP1) [8]. NOD-like receptors (NLRs) are well described pattern recognition receptors (PRRs) that bind pathogen-associated molecular patterns (PAMPs). Inflammasomes play a key role in the response to cellular stress and microbe attacks [7,9].

## 2. Gut Macrophages and Inflammation

Macrophages exposed to different stimuli can express different phenotypes. These stimuli are tissue damage, attack of pathogens, responses to stimuli originating from other immune cells, and tissue repair [10]. Based on gene expression analysis, macrophages are distinguished into two major phenotypes: the M1 proinflammatory phenotype and M2 type [10]. The M1 proinflammatory phenotype acts against yeasts, viruses, and bacteria and promotes T helper1-like immune responses. The M1 phenotype is induced by contact with pathogen-associated molecular patterns (PAMPs) and alarmins (danger-associated molecular patterns, DAMPs). M1 macrophages express molecules involved in antigen presentation and induce the production of proinflammatory mediators [10,11,12].

Monocytes that adopt the M2 phenotype are involved in the suppression of inflammation, angiogenesis, tissue remodeling, and wound healing [10,13,14].

Macrophages differentiate into an M2a subpopulation upon the influence of IL-4 and IL-13 and Th2-like immune responses. As a result, M2a macrophages are related to wound healing and promote tissue fibrosis. The M2b subpopulation is induced by immune complexes, Toll-like receptors (TLR) and/or IL-1R ligands, and bacterial lipopolysaccharide. Macrophages differentiate into an M2c subpopulation by the influence of IL-10, transforming growth factor-β (TGFβ) and glucocorticoids, Arg1, Fizz1, Ym1, CD206, and low levels of IL-12, thus playing an immunosuppressing role. The equilibrium of the healing process is dependent on the perfect functioning and cooperation of M2 subpopulations [2,10,13,14].

Gut macrophages are sentinels; therefore, they are positioned for attack in specific enteric ramparts. They mainly reside in the lamina propria (LP) of the mucosa and in the area of the muscularis externa [3]. Pro-inflammatory phenotype M1 macrophages play a pivotal role in removing the remnants of the battle from the enteric environment [2]. During the process of tissue repair, macrophages change phenotype [2]. They switch to the pro-regenerative M2 phenotype [2]. M2 macrophages produce matrix metalloproteinases and transforming growth factor β1 (TGF-β1), a pro-fibrotic factor, to recruit tissue fibroblasts and start the mucosal healing process by producing extracellular matrix (ECM) components [2]. This process is delicate, requires changes in the macrophages’ metabolism, activation of numerous transcription factors, and must be controlled to maintain a balance [2]. To maintain this balance, the role of TGF-β1 derived from regulatory T (Treg) cells is critical as it promotes limitation of inflammation via interleukin (IL)-10 [2].

In the absence of inflammation, gut macrophages show little expression of the costimulatory molecules CD40, CD80, and CD86, and are more active as phagocytes and have low expression for receptors for lipopolysaccharide (LPS) of Gram-negative bacteria. They also do not express the triggering receptor expressed on myeloid cells-1 (TREM-1) that is normally present on peripheral phagocytes. TREM-1 is a potent mediator of proinflammatory cytokines and upregulator of CD40, CD86, and CD32 [3]. Hand to hand with gut macrophages in maintaining gut mucosal homeostasis is the expansion of FoxP3+ Treg cells in the area. T-cell differentiation is dependent on resident macrophages [3].

## 3. Gut Microbiota, Mucosal Homeostasis, and Infection

The gut microbiota contain over 100 trillion microbes [6]. In humans, in descending order by abundance, the gut flora is composed of *Firmicutes*, *Bacteroides*, *Actinobacteria*, *Proteobacteria*, and *Verrucomicrobia*. The abundance and prevalence of these phyla are dependent on dietary habits, genetics, use of antibiotics, the presence of infection, and gastrointestinal tract localization [15]. In the small intestine, the most commonly abundant genera include *Lactobacillus*, *Clostridium*, *Staphylococcus*, *Streptococcus*, and *Bacteroides*. In the colon and the outer mucus layer, the mucinophilic bacteria *Akkermansia muciniphila* and *Bacteroides* species are present. Closer to the mucosa, are the aerotolerant *Proteobacteria* and *Actinobacteria*. In the region of the proximal colonic mucosa, there are facultative anaerobes like *Actinomyces* and *Enterobacteraceae.* The distal mucosa favors strict anaerobes such as *Porphyromonas*, *Anaerococcus*, *Finegoldia*, and *Peptoniphilus* [15]. The enteric mucus layer creates a friendly milieu for commensal flora to colonize and provides nutrients to the gut microbiota [15].

Gut antigen-presenting cells and microbiota cooperate to maintain the delicate balance between infection and intestinal homeostasis [5].

Intestinal macrophages and gut microbiota evolve together to provide immune tolerance to the normal microflora. In this context, intestinal macrophages in the absence of infection adopt the noninflammatory profile of gut immune quiescence [5,16]. Changes in the gut microbiota due to the use of antibiotics, alterations in host cell–microflora contact areas, and host immune system dysfunction, lead to susceptibility to pathogens [4].

In germ-free mice, peritoneal macrophages seem to lack basic functions such as chemotaxis and phagocytosis [5]. The absence of microbiota in this animal model has demonstrated intestinal lymphoid tissue defects, reduced αβ and γδ intra-epithelial lymphocytes, and reduced IgA antibodies. Th17 cells, well described as immunomodulatory effector cells, are absent in germ-free mice [4].

In mice, *Bacteroides fragilis* has proven to promote immune system maturation and is an orchestrator of the Th1/Th2 balance [4]. Upon disturbance of intestinal homeostasis, *Proteus mirabilis* interacts with monocytes to release IL-1b in a NLRP3-dependent manner [4].

*Ruminococcaceae* and *Eubacterium* of the Order *Clostridiales*, members of the intestinal flora, ferment dietary fibers and starches that cannot be decomposed by digestive enzymes to provide the host with short-chain fatty acids [17]. Short-chain fatty acids play multiple roles in gut homeostasis. They regulate the blood flow in the mucus membrane and are involved in mucus production. They also influence fluid absorption and the secretion of intestinal hormones [17].

Gut epithelial cells express receptors for short-chain fatty acids. Free fatty acid receptors 3 and 2 (FFAR3 and FFAR2) are highly expressed in the intestinal tract and cells of immune system. GPR109A/hydroxycarboxylic acid receptor 2 binds the short-chain fatty acid butyrate and is involved in intestinal Treg homeostasis. Olfactory receptor 78 is known to influence blood pressure and gut hormone secretion [17]. In experimental models using a mouse macrophage cell line, butyrate upregulates the expression of phospholipase A2 and cyclooxygenase-2, leading to an increased production of prostaglandin E2. Under LPS stimulation, butyrate also seems to increase the production of ROS and IL-1b [17]. Butyrate also influences the differentiation of monocytes to macrophages through the inhibition of histone deacetylase 3 [4].

Alterations in the balance of symbiosis between the gut microbiota and enteric epithelium is described to act in favor of protozoan invasive infections. Altered enteric flora composition has been associated with invasive infections due to *Giardia* and *Entamoeba histolytica* [15]. Communication between enteric flora and resident innate immunity components, mainly macrophages and dendritic cells, is a prerequisite for their symbiosis [3]. In the context of this symbiosis, gut macrophages can discern and recognize pathogenic bacteria from enteric flora and thus maintain a delicate balance in favor of the absence of inflammation [3]. This balance between the gut microbiota and M2 macrophages is critical for gut health and homeostasis [2]. The gut microbiota has the power to change macrophage phenotypes. Microbiota-derived metabolites, such as short-chain fatty acids (SCFAs) and Gram-negative bacterial lipopolysaccharides (LPS), exert anti-inflammatory or pro-inflammatory effects by acting on macrophages. *Bacteroides fragilis* and *Clostridia* promote the M2 phenotype [18]. Polysaccharide A of *B. fragilis* suppresses T-helper 17 responses during homeostatic gut colonization of the bacterium [3]. The prebiotic *Clostridium butyricum* pushes intestinal macrophages to induce IL-10 production thus leading to anti-inflammatory phenotype in mice [2,19]. Under inflammation, the population of gut resident macrophages can be transiently reduced [3]. It seems that the enteric microbiome induces and sets the pace of the replacement of the lost macrophage population [3]. The monocyte–macrophage infiltrate adopts a specific phenotype according to the microenvironment needs, but whether the same cell switches phenotype or cells with specific characteristics are recruited as separate cell populations needs to be further elucidated [10].

The most studied inflammasomes regarding parasitic infections are NLRP1 and NLRP3 [9]. NLRP3 is involved in infections caused by *Leishmania* spp., *Plasmodium genus*, *Trypanosoma cruzi*, *Schistosoma*, *Trichuris*, and *Fasciola hepatica* [20,21].

*Toxoplasma gondii* infection involves NLRP1. AIM2 (the absent in melanoma-2) inflammasomes are activated by host ectopic double-stranded DNA, and is involved in *Plasmodium species* infection [9,20,22].

In humans, NLRP1 is a complex formed by the proteins N-terminal pyrin domain (PYD), C-terminal caspase activating and recruitment domain (CARD), and short leucine-rich repeated domain (LRR) [9]. NLRP1B is degraded by proteolysis upon detection of pathogen-encoded enzymes from parasites. This leads to the formation of a functional inflammasome fragment [23]. The mechanism of inflammasome activation is via N-terminal degradation of NLRP1B and the release of the NLRP1B C terminus which activates caspase-1 [24].

The NLRP3 inflammasome is an intracellular innate immune sensor abundant on macrophages. It has a proteolytic action dependent on caspase-1 that leads to a cascade of interactions with the ultimate result being the induction of IL-1b and IL-18 [7,25]. Activation of the inflammasome can be either direct or indirect. Direct activation is a process that requires two signals. The first signal is NF-kB activation and leads to the up-regulation of NLRP3. The second signal leads to the assembly of NLRP3 to form an inflammasome [7]. Indirect activation is dependent on caspases-4/5 in humans and requires binding of LPS from Gram-negative bacteria [7]. Upon inflammasome activation, caspase-1 and caspases-4/5 cleave the protein gasdermin D (GSDMD), and the resulting fragment forms pores on the cell membrane. As a result, IL-1b and IL-18 passively escape extracellularly. This hyperactivation of macrophages leads to the programmed cell death called pyroptosis [7]. This mechanism of host defense is well described for intracellular parasitic infections [25]. In extracellular parasitic infections, on the other hand, NLRP3 inflammasome activation does not lead to pyroptosis, as this would be in favor of the parasite [7].

## 4. Parasite–Host Interaction

Amebiasis and amebic dysentery are caused by the eukaryotic parasite *Entamoeba histolytica*, an enteric parasite in humans. Humans are infected by the cysts via the fecal–oral route [1,26]. As a disease, it has spread worldwide and is estimated to cause 34–50 million cases of severe disease annually [26]. Of those infected, about 10% suffer from the invasive form of the disease [26]. Invasive amebic colitis is related to high mortality [23].

The protozoon exists in the environment in the form of infective cysts that can survive outside the host [26]. The cystic form also protects the parasite from the host’s gastric acids [26,27]. Upon reaching the gut lumen, excystation occurs and eight trophozoites emerge from each cyst [26,27]. The lower infective dose of amebic cysts is >10^3^ cysts [26]. After excystation, the trophozoites of the parasite may follow two pathways. They either colonize to feed on the enteric flora by phagocytosis or they cause invasive disease [26]. Invasive disease requires penetration of the colonic mucosa and binding to host enteric epithelial cells [26].

*E. histolytica* invasion presupposes host cell death which occurs after host cell–parasite contact [28]. Host cell death by the trophozoites occurs by programmed cell death, phagocytosis, and trogocytosis [28]. To invade and cause infection, the parasite has to first defeat the innate immune system soldiers. The first warrior is the mucus barrier of the colon that protects the underlying enteric epithelial cells. Mucus prevents pathogen adhesion to epithelial cells. *Entamoeba histolytica* uses Gal/GalNAc lectin to attach to host cells. Simultaneously, the parasite uses glycosidases to attack sialidase, and N-acetylgalactosamidase and N-acetylglucosaminidase to degrade the host mucin [1,7,29]. Gal/GalNAc lectin also binds to macrophages and triggers a strong pro-inflammatory response. As a result, proinflammatory cytokines like IL-1b are produced as well as nitric oxide (NO). This binding is critical to inflammasome activation [7,29].

Cysteine proteinase EhCP-A5 of *Entamoeba histolytica*, a virulent extracellularly secreted factor of the parasite, cleaves the C-terminus of MUC2 mucin. As a result, the mucus layer is damaged, creating space for the parasite to directly contact host epithelial cells and the resident macrophages of the lamina propria [7]. In response, the host induces MUC2 mucin hypersecretion from the goblet cells of the colon. The mechanism is well described. EhCP-A5 binds to the integrin receptor αvβ3 on the goblet cells of the colon. The cascade of reactions that follow via the SRC family kinases PI3K, PKCδ, and MARCKS lead to mucin exocytosis. Improper regulation of mucin exhausts goblet cells and leads to mucin depletion. In this way, the epithelium cells are exposed to invasion by the parasite [1,7]. EhCP-A5 also binds to αvβ1 integrin on macrophages that leads to NF-kB activation and induction of pro-inflammatory cytokines [7]. Exposure of host epithelial cells to the protozoon also leads to the production of human defensin 2 [1,7].

Macrophages that reside in the lamina propria are the first to orchestrate the pro-inflammatory response to *E. hystolytica* infection. The direct contact of the protozoon via binding of Gal/GalNAc lectin is the first signal to activate the NLRP3 inflammasome in macrophages [1]. The second signal is provided by EhCP-A5 of the protozoon via the following mechanism: EhCP-A5 is an integrin-binding cysteine protease on the parasite surface which binds and activates α5β1 integrin on macrophages. The cascade of reactions that follow leads to Src family kinase phosphorylation, and the subsequent pannexin-1-mediated ATP release through the pannexin-1 channel. The released ATP binds to the receptor P2×7 on macrophages. This autocrine process provides the second signal for NLRP3 activation [1,7,29]. In *Entamoeba* infection, efflux of potassium is also critical for NLRP3 activation [7].

Important weapons of the host attack against *E. histolytica* trophozoites are reactive oxygen species (ROS) and nitric oxide (NO) [26]. The protozoon has developed mechanisms to resist these oxygen species. One mechanism is phagocytosis of the host immune cells [26]. Another mechanism is the upregulation of peroxiredoxin (Prx) expression. Since the parasite lacks catalase, peroxiredoxins are critical in the reduction of hydrogen peroxide. In this manner, *Entamoeba* can be protected from the reactive nitrogen species of macrophages [30]. This requires direct cell-to-cell contact with host enteric epithelial cells. The C-terminal of Prx interacts with Toll-like receptor 4, and this interaction promotes NLRP3 inflammasome activation via the caspase-1-dependent canonical pathway [30]. Induction of IL-8, which acts as a chemoattractant to neutrophils, has detrimental effects as neutrophils are unable to kill the protozoon and provoke host tissue damage. *E. histolytica* uses this mechanism to its advantage by upregulating IL-8 production [1,28]. Trophozoites of *Entamoeba histolytica* are also capable of bypassing host immune cells. One mechanism is temporary blindness of the host’s adaptive immunity. Bound antibodies on the surface of the parasite are translocated and hidden from the complement system until different surface receptors are bound [26]. Another mechanism is using the host defense in the protozoon’s favor. The parasite secretes a protein that mimics the proinflammatory cytokine macrophage migration inhibitory factor (EhMIF) to promote invasion via the production of matrix metalloproteinases [28]. Antibodies against EhMIF seem to protect from reinfection in children [28]. The host microbiota play critical roles in *E. histolytica* colonization and invasive infection. Alterations resulting in suppression of *Clostidium, Bacteroides*, and expansion of *Bifdobacterium,* and abundance of *Prevotella copri* is related to facilitation of *Entamoeba* infection [1,7]. Experiments on mice has shown that a gut microbiome where *Prevotella copri* is dominant, shows an increase in IFNγ production by CD4+ T-cells in hosts with genetic susceptibilities. In this way, inflammation is promoted and supported [31]. On the other hand, mice colonized with *Clostridia* are protected from amebiasis via production of IL-17 and recruitment of neutrophils [32].

The extracellular flagellate parasite *Giardia* duodenalis (also known as *G. intestinalis* or *G. lamblia*) is a well-described culprit of diarrhea and enteric inflammation [33,34]. The route of infection is via ingestion of food or water contaminated with the cysts and via person-to-person contact [35]. Host–parasite interactions determine the battle outcome which is dependent on the parasite isolate [35]. *Giardia* trophozoites colonize the most proximal small intestine, but they are also found in the distal small intestine, cecum, and large intestine [36].

Not all *Giardia* trophozoites can intervene and alter host immune responses [35]. The colonization of the gut is followed by alterations in the gut milieu in favor of the parasite. To do so, the protozoon uses different mechanisms [37]. Non-specific mechanisms involve adhesion to host enteric epithelial cells by mechanical suction force via the ventral suction or adhesive disk and alterations in the zonula occludens tight junctions [37,38]. The ventral suction disk of *Giardia lamblia* is a means of attachment to the small intestine to protect the trophozoite from being carried away by enteric peristalsis and gut contents [39].

The outcome is the creation of lesions on the microvillus border, disturbance of the ionic balance, and secretory diarrhea [37]. Specific mechanisms include selective colonization of the proximal small intestine through specific interactions with lectins and annexins [39,40]. *Giardia* trophozoites attach to the host glycosylated microvillus membrane via the parasite’s lectins. Of those lectins, taglin is important for the parasite’s attachment to phospho-mannosyl residues on enterocytes of the small intestine [39,40].

The protein alpha-1 giardin of *Giardia* is expressed on the protozoon’s newly excysted trophozoites [39]. Alpha-1 giardin is a glycosaminoglycan (GAG)-binding protein, the N-terminal part of which is homologous to the conserved repeating 70 amino acid domain in annexins [39]. Alpha-1 giardin binds to the GAG heparan sulphate that is abundant on gut epithelial cells in a calcium-related manner and plays an important role in the early host–parasite interactions [39].

The gut microbiota seem to play a pivotal role not only in regulating enteric epithelium colonization but they are also implicated in disease progression [15]. In mouse models where antibiotic treatment is used, Giardia infection is more robust and persistent. This underlines the importance of balanced symbiosis of gut microbiota and host cells in *Giardia* infection [15]. *Giardia* infection alters the biofilm structure, but also the quantity, diversity, and composition of gut microbiota. These alterations act in favor of the protozoon’s colonization and are closely related to illness severity [15]. The composition of the gut microbiota plays a significant role in the manifestation of the disease. Age-related alterations in enteric flora increase susceptibility to *Giardia* infection in children less than five years old [15]. *Giardia* also provokes decreases in the thickness of the intestinal mucosa by degrading MUC2 mucin, thus facilitating the parasite’s translocation through the gastrointestinal tract [15].

Although *Giardia* duodenalis is a non-invasive parasite, both innate and adaptive immunity mechanisms are needed to control the infection and clear the pathogen. Macrophages are involved in the protozoon’s clearance [15]. Crosstalk between microbiota and resident macrophages seem to influence the induction of proinflammatory cytokines [15]. In addition, *Giardia* trophozoites use L-arginine as a nutrient. Consumption of L-arginine leads to increased synthesis of nitric oxide, a key weapon against the protozoon infection [15]. On an experimental scale, it has been proven that the NLRP3 inflammasome induces pyroptosis in macrophages in contact with the trophozoites of *Giardia* [33,34].

Interleukin 17 seems to have a protective role against *Giardia* infection [41]. In vitro experiments with human peripheral blood mononuclear cells stimulated with *G. duodenalis* showed an upregulation of IL-17 secretion [41].

The mucus layer integrity and composition are crucial in Giardia infections. Muc2-deficient mice had a higher parasite burden and suffered from a more severe illness [41].

The destruction of *Giardia*’s trophozoites is depended on complement activation [41]. In vitro experiments have shown that *Giardia*’s HSP70 immunoglobulin-binding protein, BiP, binds to TLR4. TLR2 also recognizes the parasite in vitro and leads to the phosphorylation of the MAP kinases p38 and ERK, thus leading to induction of pro-inflammatory cytokines [41]. During *Giardia lamblia* infection, upregulation of nitric oxide synthase 2 and arginase 1 in the regional macrophages has been reported [42].

The models describing the involvement of acquired immunity in *Giardia* infection in humans are mainly based on murine models and in vitro experiments. Humoral immunity is involved through the secretion of primarily IgA antibodies by the mucosal-associated lymphoid tissue. As *Giardia* antigens are T-cell-dependent antigens, cell-mediated immunity is also implicated in giardiasis [43].

There are studies revealing that the parasite disrupts human mucosal microbiota biofilms, thus rendering the epithelial cells susceptible to attacks from the gut microflora [44].

Giardia can alter the composition of the gut microbiota and the virulence of gut commensals [41].

*Giardia* infection in children less than five years old revealed that the parasite could alter the microflora’s composition, favoring *Prevotella* over *Bacteroides*. More studies should be available to elucidate the role of *Giardia* in shifting the enteric microbiota in its favor [41].

Experiments in mice have shown that colonization and proliferation of the protozoon in the small intestine leads to the expansion of aerobic commensals like *Proteobacteria* and decreases anaerobes like *Firmicutes* and *Melainabacteria* [36]. Giardia trophozoites reform the microenvironment as they are microaerophilic. The products of their metabolism favor metabolically flexible taxa that need increased oxygen tension, availability of lipids, and arginine. As a result, *Giardia* acts as a modulator of the resources available for other species to survive [36]. In this context, *Giardia* colonization provokes redox perturbations that favor facultative anaerobes, such as *Gammaproteobacteria* and *Betaproteobacteria*, and aerobes [36]. These redox alterations have an impact on the epithelial cell inflammatory responses through decreased synthesis of both host nitric oxide and interleukin-8 (IL-8) [36]. Giardia also alters the diversity and composition of the gut microbiota directly by both excreting novel lipids and influencing the bioavailability of bile acids. The protozoon creates a lipid-rich environment that favors the growth of *Acinetobacter* species [36].

Giardia trophozoites use arginine as a source of energy and compete with host cells for arginine, leading to severe malnutrition and villus shortening [36]. Reduction in the available arginine for the intestinal epithelial cells leads to a reduced ability to produce nitric oxide, a well-described inhibitor of the parasite’s replication [41].

In IL-10-deficient mice, infection with *G. muris* leads to the expansion of CD11b+, CD11c− macrophages in the colon and subsequent inflammation. No such expansion was detected in the small intestine [45].

*Giardia*-released cysteine proteases cleave Human β-Defensin 1 and α-Human Defensin in vitro in a dose-dependent manner. In this way, the parasite protects itself from antimicrobial peptides [41].

*Lactobacilli* seem to protect from *Giardia* infection in mice. They are related to a lower trophozoite burden and increase in secretory IgA antibodies and CD4+ T cells in the small intestine [41].

## 5. Conclusions

The balance between gut mucosal homeostasis and infection is dependent on crosstalk between the microbiota and resident macrophages. As far as extracellular enteric parasitic infections are concerned, a change in macrophages phenotype to a pro-inflammatory state is dependent on direct contact of the parasite with the macrophages. In the absence of inflammation, resident macrophages show little expression of the costimulatory molecules CD40, CD80, and CD86, and have increased action as phagocytes and show low expression of receptors for LPS. Gut-resident macrophages are also protagonists in healing and tissue repair post infection. The gut microbiota and enteric epithelium integrity play pivotal roles in the defense against invasive gut parasitic infections. Altered enteric flora composition facilitates invasive amebic colitis and giardiasis. Communication and equilibrium between enteric flora and resident macrophages is a prerequisite for their symbiosis. The gut microbiota has the power to change macrophage phenotype and replenish the resident macrophage niche during and post infection. The enteric flora regulates gut epithelium colonization and is also implicated in disease progression. *Giardia* disrupts the equilibrium and structure of the gut microbiota in a manner that is related to disease severity. The composition of the gut microbiota of children less than five years old also favors the establishment of the infection. Macrophages induce strong pro-inflammatory responses through inflammasome activation and secretion of proinflammatory cytokines. Direct host cell–parasite contact is needed to trigger the proinflammatory response. The enteric mucus barrier is the first hurdle for the protozoa to interact with the enteric epithelium cell and commence invasion. Parasitic lectins and mechanical force are then used to attach to host epithelial enteric cells. Resident macrophages orchestrate a pro-inflammatory response through activation of inflammasomes and induction of reactive oxygen species and nitric oxide. Parasitic infections involve NLRP1 and NLRP3 inflammasome activation. For *Entamoeba histolytica* and *Giardia* duodenalis infections, NLRP3 inflammasome activation is crucial to promote the host defenses. More studies are needed to further elucidate possible therapeutic and protective strategies against these enteric protozoan parasites’ invasive infections in humans.

## Data Availability

All data are included in the manuscript.

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
