# Peer review of "Gut Microbiota Crosstalk with Resident Macrophages and Their Role in Invasive Amebic Colitis and Giardiasis—Review"

_microorganisms, 2023, doi:10.3390/microorganisms11051203_

Round 1

Reviewer 1 Report

General comment

The review is about a very important and interest matter, the role of the gut microbiota in the regulation of the immune system and consequently their role in gut homeostasis and in the establish of gut diseases, particulary by the parasites Entamoeba histolytica and Giardia duodenalis.

Today it is known that the gut microbiota can influence the phenotype of macrophages and the production of interleukins and inflammasome activation. Indeed, the gut microbiota can influence the phenotype of macrophages through the production of short-chain fatty acids (SCFAs), which are metabolites produced by the fermentation of dietary fibers by gut bacteria. SCFAs can activate G-protein-coupled receptors (GPCRs) on macrophages, promoting an anti-inflammatory M2 phenotype. Also the gut microbiota can influence the production of interleukins, including IL-1β and IL-18, through the activation of NLRP3 inflammassome. The crosstalk between the gut microbiota and macrophages seems important in the pathogenesis of amebiasis and giardiasis. This review highlights the NLRP3 inflammasome activation in macrophages during amebiasis and their role in the pathogenesis of this parasite infection. However, few information was given in relation to Giardia duodenalis infection.

Overall, this article needs major modifications to be published in the Microorganisms.

Specific comments

Line 41: The subsection title is not adequate and not highlight the content; perhaps using subtitles like: "2. Gut macrophage and inflammation"; "3. Gut microbiota, mucosal homeostasis and infection"; "4. Parasite- host interaction"; and "5. Conclusions"

Lines 44-45: the sentence should be improved.

Lines 45-58: It seems better firstly describe what are the macrophages M1 and M2 and after the role of M2 phenotype in tissue repair. “Macrophages have been divided into two groups: classically activated M1 phenotypes, which are stimulated by IFN-γ and LPS and exert pro-inflammatory effects, and alternatively activated M2 phenotypes, which are stimulated by the IL-4 or IL-13 and perform anti-inflammatory functions, as reviewed by Tang et al. (2019)”. Also the reference number 2 not seems the adequate to support the role of the macrophage phenotypes in a general way.

Lines 67-79: About gut microbiota. What are the actions mechanisms underlying the effects of microbiota on macrophage´s phenotypes? It is known that Ruminococcaceae, Eubacterium, Clostridia, and Firmicutes were identified as the main producers of butyrate and that butyrate can exert an anti-inflammatory effect in part by suppressing the activation of NF-κB, a transcription factor that regulates the inflammatory and innate immune responses (Ohira et al., 2017). The authors should include more information’s about this issue.

Lines 88-89: It will be important include how the gut microbiota change the macrophages phenotype; perhaps “Microbiota-derived metabolites, short-chain fatty acids (SCFAs) and Gram-negative bacterial lipopolysaccharides (LPS), exert anti-inflammatory or pro-inflammatory effects by acting on macrophages.”

Line 95: and what happens in infections by the parasite Giardia duodenalis and Entamoeba histolytica? The macrophage phenotype is modified?

Lines 106-109: the sentence “Of the described inflammasomes, the NLRP3 inflammasome is an intracellular innate immune sensor abundant on macrophages. It has proteolytic action dependent on caspase-1 that leads to a cascade of interactions with ultimate result the induction of IL- 1β and IL-18 [8, 11].” should be improved.

Lines 127-131: the sentences “This degradation is either direct or indirect. In the direct pathway, anthrax lethal factor metalloprotease (LF) degrades NLRP1B to a destabilized neo-N terminus which traps LF. In accordance, Shigella flexneri E3 ligase IpaH7.8 directly degrades NLRP1B [13]. The indirect pathway seems to involve DPP8/9 inhibition which activates NLRP1B by triggering an endogenous proteasomal activity [13].” seems out of context; improve the idea or cut the information.

Lines 150-152: the sentence “Entamoeba histolytica slides over this barrier via Gal/GalNAc lectin (the parasite’s surface adhesin) and simultaneously uses glycosidases to attack sialidase, N-acetylgalactosamidase and N-acetylglucosaminidase to degrade host mucin [1, 8, 17].” should be revised

Lines 168-178: repeated information (in previous lines 153-155). Please clarify if is Gal/GalNAc or EhCP-A5 that activate the NLRP3. The authors should revise and clarify all the information of this paragraph.

Lines 172 and 173: is EhCP5? Or EhCP-A5?

Lines 179-180: in the sentence “Reactive oxygen species (ROS) and nitric oxide (NO) are important ingredients for the host immune attack to E. histolytica trophozoite invasion“ the English should be improve using scientific language.

Lines 185- 188: put together the two sentences “This requires direct cell to cell contact with host enteric epithelial cells. The C-terminal of Prx interacts with toll-like receptor 4. This interaction promotes NLRP3 inflammasome activation via caspase-1-dependent canonical  187 pathway [18].

Line 191 and line 193: change the words "fool"  and "rear"; it should be used scientific language.

Lines 198- 201: What are the mechanisms involved in this facilitation of infection? What are the bacteria factors responsible for this response?  The macrophages phenotype were modified?

Lines 202-242: In relation to Giardia duodenalis the authors presented the parasite, life cycle, and pathogenicity. However, about the gut microbiota and macrophages (the focus of the present review) few information was given and only have 1 reference (nº 5) about the influence of microbiota and macrophages in giardiasis and 2 other about inflammasome activation. How the Giardia crosstalk with microbiota and intestinal macrophages? It seems that authors don´t had no need to revise this question because all the knowledge are in the review by Fekete et al. (2021) (nº 5).

Author Response

Kalavrita, 25th April 2023

Assistant Editor, Microorganisms,

Dear Professor

We resubmit the revised version of the paper entitled “Gut microbiota crosstalk with resident macrophages and their role in invasive amebic colitis and giardiasis- opinion’’ ID (microorganisms-2308729) by Xaplanteri et al for consideration and publication in Microorganisms. All corrections are highlighted in yellow.

A point-to-point reply to all comments follows. 

All authors have agreed with the revised version of the manuscript. We declare that the entire manuscript has not been published or submitted for publication elsewhere.

Thank you for your consideration,

Sincerely yours,

Panagiota Xaplanteri, General Hospital of Eastern Achaia, Kalavrita, 25001, Greece.

Tel: 0030-2692-360130

E-mail: panagiota.xaplanteri@gmail.com

Dear Reviewers

Thank you for your comments. All suggestions have been included in this revised version of our manuscript.

All corrections are highlighted in yellow

Reviewer: 1

  • Line 41: The subsection title is not adequate and not highlight the content; perhaps using subtitles like: " Gut macrophage and inflammation"; "3. Gut microbiota, mucosal homeostasis and infection"; "4. Parasite- host interaction"; and "5. Conclusions"
    • It has been changed
  • Lines 44-45: the sentence should be improved.
    • It has been changed

  • Lines 45-58: It seems better firstly describe what are the macrophages M1 and M2 and after the role of M2 phenotype in tissue repair. “Macrophages have been divided into two groups: classically activated M1 phenotypes, which are stimulated by IFN-γ and LPS and exert pro-inflammatory effects, and alternatively activated M2 phenotypes, which are stimulated by the IL-4 or IL-13 and perform anti-inflammatory functions, as reviewed by Tang et al. (2019)”. Also the reference number 2 not seems the adequate to support the role of the macrophage phenotypes in a general way.
    • It has been changed.
  • Lines 67-79: About gut microbiota. What are the actions mechanisms underlying the effects of microbiota on macrophage´s phenotypes? It is known that Ruminococcaceae, Eubacterium, Clostridia, and Firmicutes were identified as the main producers of butyrate and that butyrate can exert an anti-inflammatory effect in part by suppressing the activation of NF-κB, a transcription factor that regulates the inflammatory and innate immune responses (Ohira et al., 2017). The authors should include more information’s about this issue.
    • Changes have been made
  • Lines 88-89: It will be important include how the gut microbiota change the macrophages phenotype; perhaps “Microbiota-derived metabolites, short-chain fatty acids (SCFAs) and Gram-negative bacterial lipopolysaccharides (LPS), exert anti-inflammatory or pro-inflammatory effects by acting on macrophages.”
    • It has been added
  • Line 95: and what happens in infections by the parasite Giardia duodenalis and Entamoeba histolytica? The macrophage phenotype is modified?
  • A comment has been added
  • Lines 106-109: the sentence “Of the described inflammasomes, the NLRP3 inflammasome is an intracellular innate immune sensor abundant on macrophages. It has proteolytic action dependent on caspase-1 that leads to a cascade of interactions with ultimate result the induction of IL- 1β and IL-18 [8, 11].” should be improved.
  • It has been rewritten
  • Lines 127-131: the sentences “This degradation is either direct or indirect. In the direct pathway, anthrax lethal factor metalloprotease (LF) degrades NLRP1B to a destabilized neo-N terminus which traps LF. In accordance, Shigella flexneri E3 ligase IpaH7.8 directly degrades NLRP1B [13]. The indirect pathway seems to involve DPP8/9 inhibition which activates NLRP1B by triggering an endogenous proteasomal activity [13].” seems out of context; improve the idea or cut the information.
  • It has been removed.
  • Lines 150-152: the sentence “Entamoeba histolytica slides over this barrier via Gal/GalNAc lectin (the parasite’s surface adhesin) and simultaneously uses glycosidases to attack sialidase, N-acetylgalactosamidase and N-acetylglucosaminidase to degrade host mucin [1, 8, 17].” should be revised
  • It has been rephrased
  • Lines 168-178: repeated information (in previous lines 153-155). Please clarify if is Gal/GalNAc or EhCP-A5 that activate the NLRP3. The authors should revise and clarify all the information of this paragraph.
  • Changes have been made
  • Lines 172 and 173: is EhCP5? Or EhCP-A5?
  • It has been corrected
  • Lines 179-180: in the sentence “Reactive oxygen species (ROS) and nitric oxide (NO) are important ingredients for the host immune attack to E. histolytica trophozoite invasion“ the English should be improve using scientific language.
  • Changes have been made
  • Lines 185- 188: put together the two sentences “This requires direct cell to cell contact with host enteric epithelial cells. The C-terminal of Prx interacts with toll-like receptor 4. This interaction promotes NLRP3 inflammasome activation via caspase-1-dependent canonical  187 pathway [18].
  • Changes have been made
  • Line 191 and line 193: change the words "fool" and "rear"; it should be used scientific language.
  • Changes have been made
  • Lines 198- 201: What are the mechanisms involved in this facilitation of infection? What are the bacteria factors responsible for this response? The macrophages phenotype were modified?
  • Additions have been made.
  • Lines 202-242: In relation to Giardia duodenalis the authors presented the parasite, life cycle, and pathogenicity. However, about the gut microbiota and macrophages (the focus of the present review) few information was given and only have 1 reference (nº 5) about the influence of microbiota and macrophages in giardiasis and 2 other about inflammasome activation. How the Giardia crosstalk with microbiota and intestinal macrophages? It seems that authors don´t had no need to revise this question because all the knowledge are in the review by Fekete et al. (2021) (nº 5).
  • The whole paragraph has been rewritten.

Reviewer 2 Report

The present review treats an interesting topic, namely the interaction betwqeen the intestinal microflora, the innate immune system and pathogens. The presentation of the relevant literature is, however, too preliminary for publication in its present form. Overall, there are only 25 references quoted in the manuscript. A succint search on pubmed with the key words "intestinal microflora" and "immune system" yields one magnitude more references.

Furthermore, the script jumps between innate immunity, microflora, and  pathogens and is therefore confusing. The "introduction" (Section 1) does not introduce the topic as a whole but merely mentions one aspect of the innate immune response.   On sections 1. and 2. follows section 5.  Where are sections 3 and 4? Would they have helped to provide more focus to the manuscript?

Last but not least, the presentation of the two intestinal pathogens Entamoeba and Giardia is too succinct to be helpful for a reader. For instance, Giardia trophozoites colonize the duodenum, whereas Entamoeba histolytica trophozoites colonize the colon. Thus, they interact with different microbial communities and different components of the host immune system. There is plenty of literature including reviews on the interaction of both pathogens with the innate immune system, which should be considered.

Author Response

Kalavrita, 25th April 2023

Assistant Editor, Microorganisms,

Dear Professor

We resubmit the revised version of the paper entitled “Gut microbiota crosstalk with resident macrophages and their role in invasive amebic colitis and giardiasis- opinion’’ ID (microorganisms-2308729) by Xaplanteri et al for consideration and publication in Microorganisms. All corrections are highlighted in yellow.

A point-to-point reply to all comments follows. 

All authors have agreed with the revised version of the manuscript. We declare that the entire manuscript has not been published or submitted for publication elsewhere.

Thank you for your consideration,

Sincerely yours,

Panagiota Xaplanteri, General Hospital of Eastern Achaia, Kalavrita, 25001, Greece.

Tel: 0030-2692-360130

E-mail: panagiota.xaplanteri@gmail.com

Dear Reviewers

Thank you for your comments. All suggestions have been included in this revised version of our manuscript.

All corrections are highlighted in yellow

Reviewer 2

  • The present review treats an interesting topic, namely the interaction between the intestinal microflora, the innate immune system and pathogens. The presentation of the relevant literature is, however, too preliminary for publication in its present form. Overall, there are only 25 references quoted in the manuscript. A succint search on pubmed with the key words "intestinal microflora" and "immune system" yields one magnitude more references.
    • The whole manuscript has been rewritten. Interaction of intestinal microflora and immune system has been restricted mainly to macrophages and their role in gut homeostasis and infection.
  • Furthermore, the script jumps between innate immunity, microflora, and pathogens and is therefore confusing. The "introduction" (Section 1) does not introduce the topic as a whole but merely mentions one aspect of the innate immune response.   On sections 1. and 2. follows section 5.  Where are sections 3 and 4? Would they have helped to provide more focus to the manuscript?
    • Additions has been made in introduction section. Sections 3 and 4 have been added.
  • Last but not least, the presentation of the two intestinal pathogens Entamoeba and Giardia is too succinct to be helpful for a reader. For instance, Giardia trophozoites colonize the duodenum, whereas Entamoeba histolytica trophozoites colonize the colon. Thus, they interact with different microbial communities and different components of the host immune system. There is plenty of literature including reviews on the interaction of both pathogens with the innate immune system, which should be considered.
    • The whole manuscript has been rewritten.

Round 2

Reviewer 2 Report

Compared to the first version, the revies version of the present manuscript has been clearly improved. There are almost twice as much references, and  the script has a better structure. Hovever, the focus of the script can be improved. for instance, the authors mention the inflammasomes NLRP1 and 3 in the abstract and in the text l. 128, but describe these complexes in more detail only in ll.157ff. This paragraph should be moved to the "Introduction". Here, various abbreviations used in the script could also be explained.

Some details to consider:

l. 180. Do all parasitic infections involve the inflammasomes NLRP1 and 3 (i.e. also helminthic infections), or only the two protozoal parasites mentioned in this review?

l. 181f. Do you want to say: "NLRP1 is a complex formed by the proteins PYD, CARD, NACHT, and LRR (explain!).."?

What about NLRP2?

L312 f. The destruction of Giardia trophozoites (not "Giardia's killing..) ist not only dependend on complement activation. There is also some evidence that lytic antibodies targeting the variant surface proteins may be responsible. A suitable original reference may be included. The reference 38 quoted in this paragraph is a review.

Author Response

 Kalavrita, 29th April 2023

Assistant Editor, Microorganisms,

Dear Professor

We resubmit the revised version (2308729-revision 3) of the paper entitled “Gut microbiota crosstalk with resident macrophages and their role in invasive amebic colitis and giardiasis- review’’ ID (microorganisms-2308729) by Xaplanteri et al for consideration and publication in Microorganisms. All corrections are highlighted in green.

A point-to-point reply to all comments follows.

All authors have agreed with the revised version of the manuscript. We declare that the entire manuscript has not been published or submitted for publication elsewhere.

Thank you for your consideration,

Sincerely yours,

Panagiota Xaplanteri, General Hospital of Eastern Achaia, Kalavrita, 25001, Greece.

Tel: 0030-2692-360130

E-mail: panagiota.xaplanteri@gmail.com

Dear Reviewers

Thank you for your comments. All suggestions have been included in this revised version of our manuscript.

All corrections are highlighted in green

Reviewer: 2

1) Compared to the first version, the version of the present manuscript has been clearly improved. There are almost twice as much references, and  the script has a better structure. However, the focus of the script can be improved. for instance, the authors mention the inflammasomes NLRP1 and 3 in the abstract and in the text l. 128, but describe these complexes in more detail only in ll.157ff. This paragraph should be moved to the "Introduction". Here, various abbreviations used in the script could also be explained.

  • The paragraph has been removed to Introduction. All abbreviations have been explained.

2) l. 180. Do all parasitic infections involve the inflammasomes NLRP1 and 3 (i.e. also helminthic infections), or only the two protozoal parasites mentioned in this review?

  • Comments have been added.

3) l. 181f. Do you want to say: "NLRP1 is a complex formed by the proteins PYD, CARD, NACHT, and LRR (explain!).."? What about NLRP2?

  • It has been changed. Comments have been added

4) L312 f. The destruction of Giardia trophozoites (not "Giardia's killing..) ist not only dependend on complement activation. There is also some evidence that lytic antibodies targeting the variant surface proteins may be responsible. A suitable original reference may be included. The reference 38 quoted in this paragraph is a review.

  • It has been changed.